# Life Cycle Assessment of the Cultivation Processes for the Main Vegetable Crops in Southern Egypt

**Mostafa Abdelkader** [1,*], **Meisam Zargar** [2], **Kheda Magomed-Salihovna Murtazova** [3] **and Magomed Ramzanovich Nakhaev** [4]

1   Horticulture Department, Faculty of Agriculture, Sohag University, Sohag 82524, Egypt
2   Department of Agrobiotechnology, Agricultural-Technological Institute, RUDN University, 117198 Moscow, Russia; zargar_m@pfur.ru
3   Engineering Center of Carbon, Kadyrov Chechen State University, 364024 Grozny, Russia; fu.ggni@mail.ru
4   Applied Mathematics and Computer Technology, Kadyrov Chechen State University, 364024 Grozny, Russia; mr-nakhaev@mail.ru
*   Correspondence: m.abdelkader@agr.sohag.edu.eg

**Abstract:** Due to the increasing concern about climate change and environmental sustainability, the investigation of energy consumption represents a very intriguing and undeniable subject. This study was directed to investigate energy footprints, greenhouse gas (GHG) emissions and life cycle assessment (LCA) of the main vegetable crops cultivated under open field conditions in southern Egypt. Potato production required the maximum energy amount (112.3 GJ/ha) compared to 76 GJ and 96 GJ for onion and tomato, respectively. Based on energy indices, potato gave (energy ratio > 1; energy productivity > 1; energy profitability > 1; net energy > 0), while onion and tomato production shared the same indicators (energy ratio < 1; energy productivity > 1; energy profitability < 0; net energy < 0). However, GHG emissions generated for producing one ton of potato tubers registered the least amount by 76.0 kg $CO_2$ eq. The same GHG amount was produced by 834 kg of onion bulbs and 940.6 kg of tomato fruits. The emission rates were more a consequence of diesel, followed by inorganic fertilizer and manure. In addition to carbon emissions, every production process causes several other environmental problems, thus a comprehensive analysis of environmental impact categories is required. The openLCA program performed LCA and ten impact categories were considered to transform the inventory data into several indicators. Producing one ton of potato tubers has the least footprint on the environment and the ecosystem, such as global warming (GW)— 238.8 kg $CO_2$ eq. $t^{-1}$; human toxicity (HT)—288.3 kg 1,4-DB eq. $t^{-1}$; fresh water aquatic ecotoxicity (FAEF)—160.44 kg 1,4-DB eq. $t^{-1}$; marine aquatic ecotoxicity (MAET)—365,636 kg 1,4-DB eq. $t^{-1}$; and terrestrial ecotoxicity (TE)—1.18 kg 1,4-DB eq. $t^{-1}$. The analyses indicated that machinery and diesel fuel had the highest impact on all the studied categories.

**Keywords:** energy balance; GHGs; LCA; onion; potato; tomato; sustainability

## 1. Introduction

The long-term struggle with nature led to critical degradation of the agricultural environment, as evidenced by the altering erosion of fertile lands, deforestation activities, chemical materials and air pollutants. On the other hand, the extraordinary increasing population means an increase in food and water consumption and, consequently, energy usage, leading to severe economic and environmental problems [1]. Currently, global warming and climate change are the most crucial challenges facing the world. It represents a fundamental danger to livelihoods, ecosystems, water resources, infrastructure and the global economy. Policies makers collaborate with scientists to control global GHG emissions, significantly reduce the extent of climate change in the future and avoid its expected terrible impacts that would weaken developmental gains [2].

Climate change has become a pressing alarm for Egypt, which recorded one of the most severe heat waves in August 2021 with more than 40 °C. Egypt developed five clear strategic goals that co-exist with the strategy for sustainable development, "Egypt's Vision 2030", which includes lowering GHG emissions. In 2019, Egypt generated 246.64 million tons of carbon with a slight reduction compared to 2018 which saw 251.46 million tons generated, representing 0.68% of global production at 2.46 tons of carbon per capita [3]. Despite Egypt having a low fingerprint of $CO_2$ emissions per capita compared to the USA (15.5 tons), Russia (11.4 tons) and China (7.4 tons) [4], reducing GHGs by heading towards renewable energy resources and energy efficiency is consistent with Egypt's long-term development goals [5]. The Climate Change Risk Management Program asserts that Egypt is moving towards a less GHG-intensive path mainly by becoming a more energy-efficient economy and by increasing the usage of its sizeable renewable energy potential.

Agriculture is still a crucial pillar of the Egyptian economy. The Egyptian economy depends heavily on the agricultural sector to save food, fiber and other products. It provides livelihoods for more than half the population and employs 27.5% of the labor force [6]. The sector formed 12% of the Egyptian economy in 2015. The current primary goal is enhancing GHG inventory data, sources and capacity building to design sustainable systems and mitigation actions that promote reforms in policies and investments that indirectly decrease vulnerability to climate change [3].

Understanding the behavior of energy consumption and greenhouse gas (GHGs) emissions in crops production is the key to deeply analyzing the structure of the agricultural environment [7] due to the fact that energy is the impulsive power of existence and is necessitated for all agricultural production systems [8]. Agriculture activities are considered one of the principal global emitters of GHGs. Fertilizers, pesticides, machinery, manure and irrigation water are the most sources of GHGs in farming [9]. The production process is the main category for evaluating energy consumption and GHG emissions from agricultural systems [10]. In the studied area that belongs to arid zones conditions, which cover 40% of the land area in the planet [11,12], energy balances, GHG emissions and life cycle assessment (LCA) in agricultural production systems have never been studied before.

This work aims to measure energy use efficiency and GHG emissions from the main vegetable crops (onion, potato and tomato) in Sohag Governorate, southern Egypt, and then computes the potential impact on the ecosystem. This study also aims to define the impact generated by the cultivation of the three crops in the referred area and to provide sector operators with the potential impact of each crop to achieve sustainability in this region. On the other hand, the aim is to make consumers aware of how much impact one unit of the product consumes and generates. These data can provide essential information for pursuing low-carbon agriculture and adjusting vegetable production systems in southern Egypt.

## 2. Materials and Methods

### 2.1. Crop Selection

This study was conducted in Sohag Governorate, located in the southern region of Egypt, and covered a stretch of the Nile Valley. The majority of residents (79%) of this governorate lives in the rural areas and work in the agricultural sector [13]. Onion, potato and tomato are essential cash crops in Egypt that generate high income [14]. Tomato is ranked as the first crop among vegetables in the cultivated area and total production with approximately 160 thousand hectares representing 28% of the total vegetable area, producing 6.75 million tons in 2019. Egypt is the fifth-biggest tomato producer globally after China, India, Turkey and the USA [15]. Tomato fruits provide beneficial effects through their high content of minerals such as potassium and antioxidants such as vitamin C, vitamin A, lycopene and tocopherol [16]. Tomato originated in America and was then introduced to Europe and Mediterranean countries at the beginning of the sixteenth century [17]. According to the environmental conditions, Egypt has the ability to produce tomatoes in all governorates and all seasons, which leads to the availability of tomatoes all year round. In Sothern Egypt, tomato is often cultivated in September and the crop is then harvested

from January to March. Hybrids such as super strain B, G.S 12 and C. L 150 are usually cultivated in this region.

Onion is considered one of the most cultivated vegetable crops after tomato, with an estimated 5.2 million hectares of the harvested area, producing 100 million tons worldwide [15]. Onion is an important export crop in the Egyptian trade, with more than 65 thousand hectares of cultivated area and 120.25% as a self-sufficiency rate of onion production. Saudi Arabia, Russia and the Netherlands came to the forefront of importing Egyptian onions [18]. Onions are rich in phosphorus, calcium and carbohydrates in addition to playing a role in preventing heart disease [19]. Onion is a winter-season crop cultivated in September and the bulb is harvested in April. Giza 6 Mohassan, Giza 20 and Shandaweel 1 represent all cultivars sowed in the Sohag Governorate. Potato (*Solanum tuberosum* L.) belongs to the family Solanaceae, which originated in Southern America [20]. Potato consumption provides the human body with essential nutrients that include carbohydrates, protein, vitamin C, vitamin B6, magnesium, potassium and fiber, making it a healthy nutrient with low calories [21]. Additionally, potato tuber is used as an industrial crop to produce starch [22].

Potatoes are Egypt's largest horticultural export, with 734 thousand tons exported in 2018 to European countries such as Greece, Italy, Germany and England [13]. The cultivated area of potatoes in Egypt is about 175.2 thousand hectares, with a production of 5.1 million tons [15], with a self-sufficiency rate reached for potatoes 111.4% in 2018 [23]. Potato in this region is cultivated in October and its crop is given in February, and it is basically produced for early export to European countries. Spunta and Diamant are the most cultivated varieties. For receiving a more objective perspective, we present the first calculation of the environmental impacts of the main vegetable cultivation systems in this region using a life cycle assessment method (LCA). We specify standard measurable values of resource depletion, acidification, eutrophication hazards and global warming potential. This helps to identify which crop is in the critical status of intervention to mitigate the impacts.

*2.2. Data Collection*

In this step, all inputs and outputs parameters were gathered, specified and separated into two groups [24]. The first group contains data from the foreground system representing how many materials and energy carriers were used in the farming activities. The second group contains data from the background system linked to the extraction of raw materials and production [25]. A number of agricultural inputs used in farming, including agricultural machinery, fertilizers, human labor, fuel and obtained yield, were collected through surveys [26]. Data were collected from the producers by using a face-to-face questionnaire. The collected data belonged to the 2020/2021 seasons. Sample farms were randomly chosen using a stratified random sampling technique. The permissible sample size error was defined as 5% and the sample size was estimated as 120 farmers [27,28].

The collected information from the farmers included crop type, sowing date for each crop, the number of seeds used and the cultivated variety, type and rate of fertilizers used, amount of physical work for the whole crop period, number of pesticides and fungicides used, fuel consumption and machinery used and yield of the crop.

Farmers in southern Egypt often irrigate their crops using furrow and flood methods. Due to the difficulty of perfectly determining the amount of irrigation water for these methods, CROPWAT 8.0 program (CROPWAT is a decision support tool developed by FAO, Rome, Italy) was used to accurately compute the total water amount per hectare ($m^3$). Metrological information such as stations, station longitude and latitude, altitude, temperature, humidity, wind speed and solar radiation were included in the model using CLIMWAT 2.0 (a climatic database to be used in combination with the computer program CROPWAT, Rome, Italy) to calculate reference evapotranspiration in CROPWAT software. Crop characteristics include the length of each developmental stage (initial, mid-season, late), depletion coefficient, root depth, crop coefficient and yield response factor. Soil types

and planting dates of each crop are also adjusted in the program [29]. During surface (furrow) irrigation, low irrigation efficiency is caused by water losses by runoff, evaporation from water in the furrow channels, evaporation from the soil surface and percolation below the root zone. Runoff losses can be substantial if not reused. Furrow irrigation efficiency is between 45–65% [30]. In our case, irrigation efficiency in CROPWAT was set at 60%.

### 2.3. Data Calculation and Method Conversion

The underlying principle is the calculation of total energy for the production process by converting the individual inputs and outputs into units of mega joule (e.g., ha) and evaluating based on energy consumed and GHGs emitted per unit by the adding the partial energies of each input or output referenced to the process of crop production. Energy inputs were human labor, diesel fuel, machinery, farmyard manure, water irrigation, synthetic fertilizers (nitrogen, phosphorous and potassium) and chemicals (herbicide, fungicide and insecticide). The average input use was recorded to assess inputs energy (MJ/ha). All inputs were calculated per hectare basis. The amount of each input is multiplied by their relevant energy equivalent coefficient [28]. The energy equivalent coefficient showed the amount of generated energy from one unit of input such as 1.96 MJ/h human labor [31,32], 62.7 MJ/h machinery [33], 43.5 MJ/l diesel, 92.1 MJ/kg nitrogen, 13.4 MJ/kg phosphorus, 9.2 MJ/kg potassium [32], 0.3 MJ/tonne manure [34], 102.1 MJ/kg chemicals [35], 1.02 MJ/m$^3$ water [36,37], 1 MJ/kg onion [19] and tomato [34] seeds and 3.6 MJ/kg potato seeds [38]. Output energy was calculated by multiplying the total production of the specific crop by its energy equivalent, such as 1.85 MJ/kg onion bulbs [25], 3.6 MJ/kg potato tubers [38] and 0.8 MJ/kg tomato fruits [34].

Based on obtained energy equivalents, we evaluate the performance of energy utilized in the production of onion, potato, and tomato by creating relevant indices: energy ratio, energy productivity, specific energy, energy profitability, and net energy gain. The energy ratio (ER), also called Energy Use Efficiency (EUE), is the ratio of outputs energy to inputs energy in production factors. This index indicates the influence of inputs expressed in the energy unit of energy output. ER can be ameliorated in the production processes by reducing the sequestered energy of inputs and/or increasing crop yields [32]. Energy productivity (EP) measures the ratio of produced yield of the unit (hectare) and the total consumed energies for this process and expressed in kg/MJ. ER (EUE) and EP were calculated [33] as follows:

$$ER = \frac{\text{outputs energy } \left( MJ \cdot ha^{-1} \right)}{\text{inputs energy } \left( MJ \cdot ha^{-1} \right)} \tag{1}$$

$$EP \left( kg \cdot MJ^{-1} \right) = \frac{\text{Crop yield } (kg \cdot ha^{-1})}{\text{Consumed energy } (MJ \cdot ha^{-1})} \tag{2}$$

Specific energy (SE) measures the inputs' energy amount required for producing one unit of output and expressed in MJ/kg. SE was calculated [39], it is also the inverse of (EP).

$$SE \left( MJ \cdot kg^{-1} \right) = \frac{\text{Consumed energy } (MJ \cdot ha^{-1})}{\text{total crop yield } (kg \cdot ha^{-1})} = \frac{1}{EP} \tag{3}$$

Net energy (NE) is the difference between the energy used in producing output to the total energy necessitated in producing output (inputs), expressed in MJ/ha:

$$NE \left( MJ \cdot ha^{-1} \right) = \text{Total produced energy } \left( MJ \cdot ha^{-1} \right) - \text{Total consumed energy } \left( MJ \cdot ha^{-1} \right) \tag{4}$$

Energy profitability (EPB) is a relative number that measures the efficiency of the production process, determined as follows [25]:

$$EPB = \frac{NE \left(MJ \cdot ha^{-1}\right)}{Consumed\ energy\ \left(MJ \cdot ha^{-1}\right)} \tag{5}$$

The distribution of consumed energy was divided to direct (labor, diesel and water) and indirect (machines, fertilizers, pesticides, manure and seeds). It also was sorted as renewable (labor, manure, seeds and water) and non-renewable (machinery, diesel, chemicals and fertilizers).

### 2.4. Greenhouse Gas Emissions (GHGs)

Carbon dioxide ($CO_2$) is the primary source of global warming [40]. $CO_2$ emission coefficients were applied to quantify GHG emissions from the different cultivation processes. The amount of each input during the cultivation was multiplied with respective emission coefficients. Applied coefficients of carbon emissions of inputs in this research were as follows: human labor (0.11 kg $CO_2$ eq. $h^{-1}$) according to Yan et al., 2014 [41]; diesel (2.76 kg $CO_2$ eq. $L^{-1}$) and machinery (0.071 kg $CO_2$ eq. $MJ^{-1}$) according to Dyer and Desjardins, 2006 [42]; chemicals (5.1 kg $CO_2$ eq. $kg^{-1}$), nitrogen fertilizers (1.3 kg $CO_2$ eq. $kg^{-1}$), phosphorus (0.2 kg $CO_2$ eq. $kg^{-1}$) and potassium (0.2 kg $CO_2$ eq. $kg^{-1}$) according to Lal, 2004 [43]; and manure (0.126 kg $CO_2$ eq. $kg^{-1}$) according to Wang et al., 2020 [44]. According to Ilahi et al., 2019 [45], the amount of each input used during the cultivation process was multiplied with equivalent emission coefficients and GHG emissions (kg $CO_2$ equivalent) per unit area (hectare) were calculated. Then, the results were tabulated by taking into consideration the inputs and input–output values of each studied crop.

### 2.5. Life Cycle Assessment

Life cycle assessment (LCA) is used to investigate the impact of a product on the environment [46]. LCA is utilized in LCA to set up the relation between the elementary flows inventory of the product system and its potential impacts on the environment and the ecosystem. Selection of impact categories and classification are the first two mandatory steps of LCA [47]. The "cradle-to-farm gate" system boundary was selected for this study because such a system boundary covered all stages associated with LCA of production processes. The term "cradle" refers to the upstream processes such as the production of fertilizers, chemicals and other auxiliaries applied within the system boundary, while the term "farm gate" refers to the harvesting stage. Figure 1 shows the adopted system boundary. Machinery, irrigation, fuels, fertilizers and chemicals are within the scope of the study. The openLCA 1.10.3 software (open-source and free software, Green Delta, GmbH, Berlin, Germany) was employed to compute LCIA and obtain the impact assessment results. Characterization factors were extracted from agriballyse database, which is incorporated into the openLCA software. Inventory data were imported to the openLCA software to perform LCA. The CML2 baseline V3.04/EU25 method was applied, while ReCiPe 2016 Midpoint (H) was applied for the impact assessment method.

Regarding the system boundary, a study "from cradle-to-farm gate" (Figure 1) was carried out in agreement with ISO 14040 [48]. In fact, the LCA study does not often cover the whole production process but can be determined to be a part of it [49]. In harmony with the guidelines of ISO 14040 [48] and ISO 14044 [50], selecting impact categories and categorization is a mandatory step in the life cycle impact assessment [51]. The selection of a proper system boundary helps to precisely estimate emissions from direct and indirect inputs and even downstream processes [52]. The mandatory steps achieve the aims of this work. In such a way, ten impact categories were selected as the most influential impact categories and the data were analyzed to calculate the index of the determined categories. The functional unit (a ton of production) was specified [53] and the goal was adjusted to

measure the environmental impacts of vegetable cultivation processes in southern Egypt, determine the highest input contribution and establish potential improvements to decrease such negative impacts.

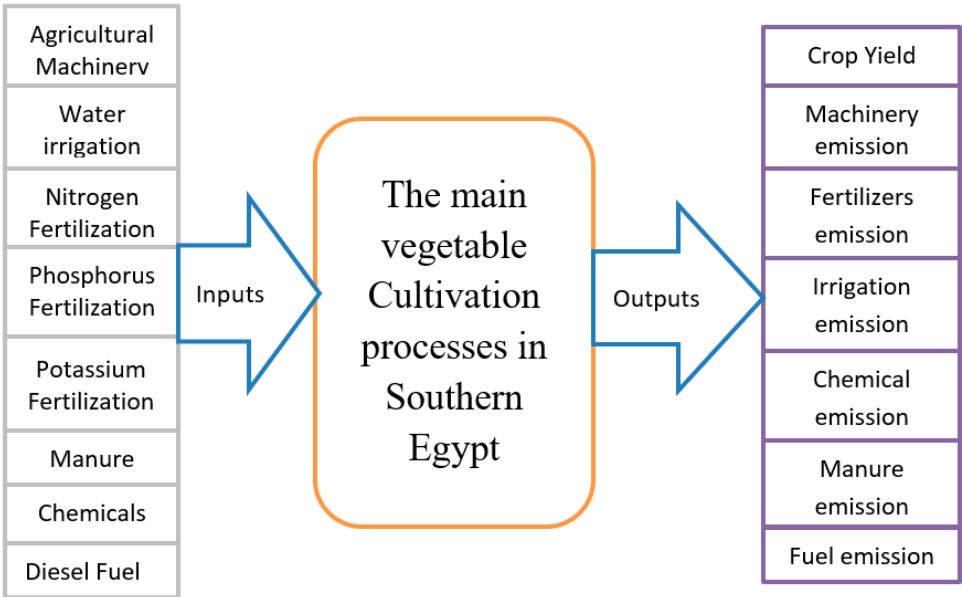

**Figure 1.** System boundary of cultivation processes (onion, potato and tomato).

## 3. Results and Discussion

### 3.1. Inputs, Outputs and Energy Equivalents

The number of inputs and outputs of production processes for the three studied crops (onion, potato and tomato) and energy equivalents (MJ/ton) are given in Tables 1 and 2. The results showed significant differences among the three crops. Compared to onion and potato, tomato is one of the most physically demanding (1474 h/ha) crops in the studied region.

**Table 1.** Number of inputs and outputs of production processes for onion, potato and tomato.

| Process | Onion | Potato | Tomato | $LSD_{0.05}$ |
|---|---|---|---|---|
| Human labor (h/ha) | $1253 \pm 32.1$ | $1224 \pm 57.2$ | $1474 \pm 28$ | 57.3 |
| Machinery (h/ha) | $81.4 \pm 3.8$ | $72.2 \pm 4.7$ | $88.0 \pm 3.9$ | 3.2 |
| Diesel (L/ha) | $366.3 \pm 12.8$ | $327.4 \pm 10.4$ | $422.6 \pm 15.6$ | 14.4 |
| Nitrogen (Kg/ha) | $380.5 \pm 20.4$ | $576.0 \pm 29.2$ | $513.8 \pm 18.8$ | 39.1 |
| Phosphorus (Kg/ha) | $226 \pm 5.7$ | $312 \pm 18.6$ | $284.2 \pm 37.8$ | 21.2 |
| Potassium (Kg/ha) | $97.6 \pm 7.6$ | $124.5 \pm 19.4$ | $199.8 \pm 13.2$ | 23.8 |
| Manure (t/ha) | $6.90 \pm 0.43$ | $7.91 \pm 0.45$ | $8.01 \pm 0.69$ | 0.4 |
| Chemicals (kg/ha) | $39.78 \pm 1.63$ | $33.10 \pm 2.40$ | $48.69 \pm 2.80$ | 4.1 |
| Water ($m^3$/ha) | $7441 \pm 184$ | $7783 \pm 248$ | $8762 \pm 226$ | 371 |
| Seed (kg/ha) | $25.4 \pm 1.49$ | $5304.2 \pm 438$ | $0.78 \pm 0.05$ | 425 |
| Yield (ton/ha) | $35.7 \pm 2.6$ | $44.2 \pm 4.2$ | $48.4 \pm 3.8$ | 5.7 |

The three main mineral elements in crop nutrition are nitrogen (N), phosphorus (P) and potassium (K). Together they make up the trio known as NPK. In accordance with FAO data, the consumption of the trio elements dramatically increased by more than 550% in the last 60 years in Egypt (Figure 2). Despite fertilizers providing crops with essential nutrients for producing more food, non-controlled application of fertilizers leads to more release of greenhouse gas emissions and eutrophication.

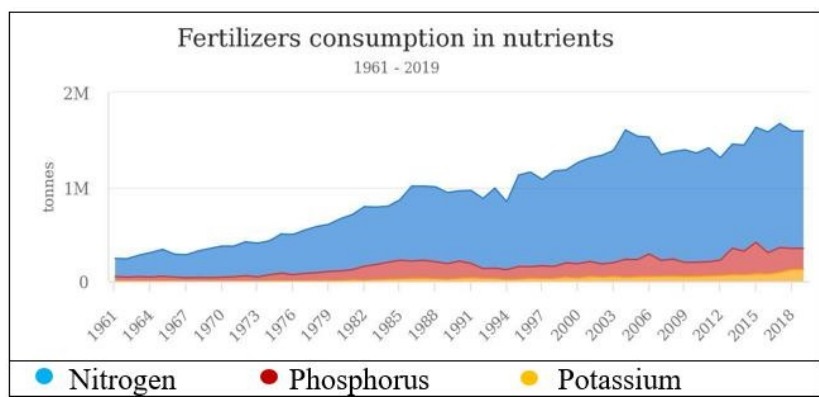

**Figure 2.** The total consumed fertilizers for the main three nutrients in Egypt from 1961 to 2019 according to FAO data [54].

The highest applied amount of fertilizers (1012.5 kg/ha) was observed in potato farms with more than 65% contribution of mineral nitrogen fertilizer. Onion and tomato fields were applied by 704.1 and 997.8 kg/ha with 60% nitrogen fertilization. Average machinery and burned diesel fuel used in tomato fields were 88.0 h/ha and 422.6 L/ha, respectively, while potatoes fields required the least hours (1124 h) of manual work, 72.2 h for machinery and 327 L of diesel per hectare. The number of organic fertilizers used decreased by 15% on onion fields compared to potato and tomato farms which consumed about 8.0 tons/ha. Again, in the same table, tomatoes consumed the highest irrigation water rate (8762 m$^3$/ha) and chemical materials (48.7 kg/ha). These amounts significantly decreased to 7783 m$^3$ and 33.1 kg in potatoes. Potato and tomato yields are significantly no different with more than 44 tons/ha compared to 35.7 of onion bulbs.

**Table 2.** Energy equivalent (MJ/ton) of inputs of the cultivation processes of vegetable crops.

| Inputs | Onion | % | Potato | % | Tomato | % |
|---|---|---|---|---|---|---|
| Human labor | 68.8 | 3.2 | 54.3 | 2.1 | 59.8 | 3.0 |
| Machinery | 143.0 | 6.7 | 102.5 | 4.0 | 114.1 | 5.7 |
| Diesel | 446.4 | 20.9 | 322.5 | 12.7 | 380.0 | 19.1 |
| Nitrogen | 981.6 | 46.0 | 1200.3 | 47.2 | 978.5 | 49.3 |
| Phosphorus | 84.6 | 4.0 | 94.6 | 3.7 | 78.7 | 4.0 |
| Potassium | 24.9 | 1.2 | 25.9 | 1.0 | 38.0 | 1.9 |
| Manure | 58.0 | 2.7 | 53.7 | 2.1 | 49.7 | 2.5 |
| Chemicals | 113.8 | 5.3 | 76.5 | 3.0 | 102.8 | 5.2 |
| Water | 212.6 | 10.0 | 179.8 | 7.1 | 184.8 | 9.3 |
| Seed | 0.71 | 0.0 | 432.4 | 17.0 | 0.016 | 0.0 |
| Energy/ton | 2134.5 | | 2542.5 | 100.0 | 1986.4 | |
| Energy/ha | 76,202.3 | | 112,276.9 | | 96,061.9 | |

Based on the energy equivalences, we can notice that producing one ton of potato consumed the maximum number of inputs (2542.5 MJ), which increased more than 400 MJ than onion and tomato. The highest consumed item was nitrogen fertilizer, which registered 46.0, 47.2 and 49.3% of the total consumed energy for producing one ton of potato, onion and tomato. Energy eq. per ton in tomato farms gave the least mega joules (1986.4) compared to onion and potato.

The energy used from diesel came in second place by 21% and 19% in onion and tomato fields. While in potato, true seed energy ranked second with 432.4 MJ/ton representing 17%, and diesel energy representing 12.7%. Energy equivalent shared from seeds of onion and tomato represents the minimum amount (less than 0.6 MJ/ton). The least chemical rate in potato produced 80 MJ/ton represents 3.0% of total energy eq. At the same time, this percentage increased to 5.25 in onion and tomato. Khoshnevisan et al. [55] reported that input energy was calculated at 3644 MJ ton$^{-1}$ of potato. While Jadidi et al. [56] showed

that tomato production consumed 65,238.9 MJ/ha, of which mineral fertilizers represented 51%. Manure, seeds and diesel were observed as the most energy-consuming inputs in potato fields by 49, 24 and 12% [57]. Potato seeds' energy ranked second in the potato cropping system. This is due to the fact that potato cultivation needs a considerable amount of potato tubers per hectare. Additionally, potato tuber (seeds) produces 3.6 MJ/kg as energy equivalent.

Collected data of each input per hectare (Figure 3) showed that the average energy amount realized from nitrogen fertilization represented the highest energy input with more than 35 GJ/ha. Diesel occupied the second position with more than 14 GJ/ha. All other inputs produced less than 10 GJ/ha and potassium fertilization generated the least amount (<4 GJ/ha).

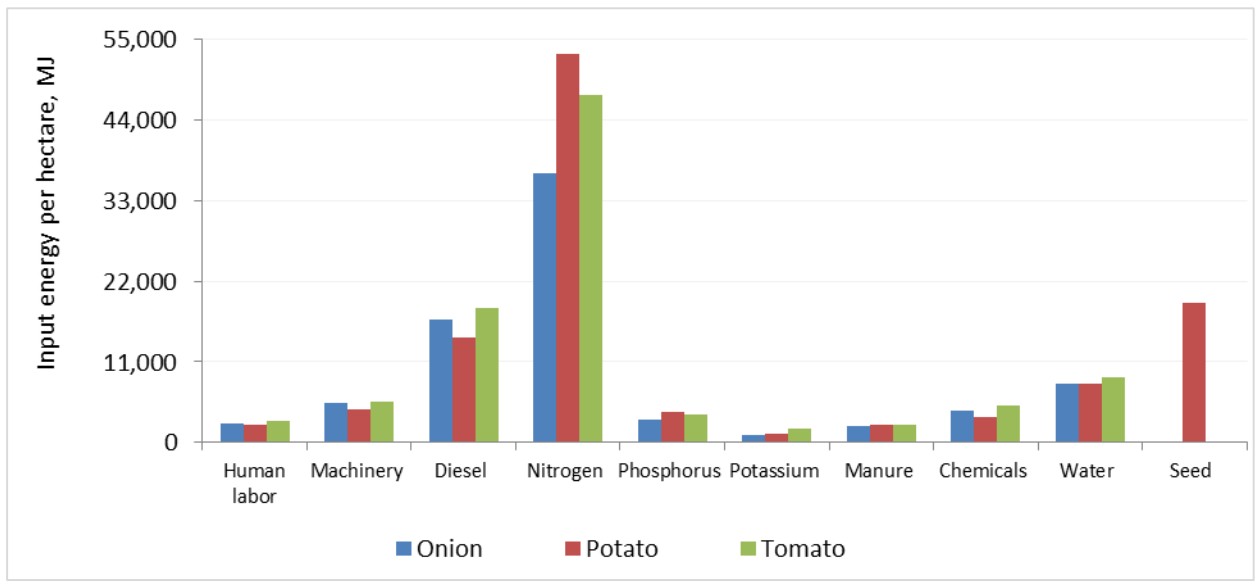

**Figure 3.** Amount of consumed energy used for one-hectare cultivation.

### 3.2. Energy Forms and Measurement Indices

Table 3 displays indices of consumed and emitted energy (MJ/ha) for the studied vegetable crops in southern Egypt. During production, potatoes consumed (102 GJ) and generated (158 GJ) the highest energy rates compared to onion and tomato. Generally, in potato fields, the generated energy is estimated to be one and a half times more compared to the consumed energy, and vice versa in onion and tomato farms, whereas consumed energy (76 GJ and 95 GJ) was more than the produced (66 and 88 GJ), respectively.

As identified, ER of potato farms (1.42) was significantly higher than that calculated in onion and tomato farms (0.87 and 0.40), respectively. Since energy productivity measures the ratio of hectare yield divided by consumed energy, we found that potato fields have the minimum EP (0.39 kg/MJ) compared to onion (0.47 kg/MJ) and tomatoes (0.50 kg/MJ). Specific energy measures the amount of consumed energy required to produce one kg of the specific crop. In our case, the minor SE was obtained from tomato farms (1.98 MJ/kg), followed by onion (2.13 MJ/kg) and potato (2.54 MJ/kg).

Considerable differences were observed in the amount of net energy among the studied crops. NE in potato fields was more than 45 GJ. On the contrary, negative NE values were calculated from onion and tomato fields, showing 10 and 57 GJ for one hectare. The distribution of consumed energy as direct (labor, diesel and water), indirect (machines, fertilizers, pesticides, manure and seeds), renewable (labor, manure, seeds and water) and non-renewable (machinery, diesel, chemicals and fertilizers) forms is shown in Table 3. It can be illustrated that 34, 22 and 31% of total energy input resulted from direct energy, and 16, 28 and 15% from renewable energy for onion, potato and tomato, respectively

(Figure 4). The high percentage of non-renewable energy consumed in vegetable production in this region revealed that these production processes depend primarily on fossil fuels. Our finding agrees with Mohammadi et al. 2008 [33], with the research results confirming that direct, indirect, renewable and non-renewable accounted for 18, 82, 26 and 74% in potato production. Hamedani et al. [58] found similar results of potato production's direct and indirect energy use percentages. In other studies, ER located between 0.5–0.8, and EP ranged from 0.7 to 0.9 kg/MJ [34,59,60]. This inefficiency is due to the conventional farming systems of vegetable crops and the shortage of management of consumed inputs, especially synthetic fertilizers.

**Table 3.** Energy forms and indices of production processes of vegetable crops.

| Indicator | Unit | Onion | Potato | Tomato | Mean |
|---|---|---|---|---|---|
| Total inputs (MJ) | MJ·ha$^{-1}$ | 76,202.3 | 112,276.9 | 96,061.9 | 94,847.03 |
| Total output (MJ) | MJ·ha$^{-1}$ | 66,029.6 | 158,970.0 | 38,687.3 | 87,895.64 |
| Yield (kg/ha) | kg·ha$^{-1}$ | 35,700 | 44,200 | 48,400 | 42,766.67 |
| Energy productivity | kg·MJ$^{-1}$ | 0.47 | 0.39 | 0.50 | 0.46 |
| Specific energy | MJ·kg$^{-1}$ | 2.1345 | 2.5402 | 1.9848 | 2.22 |
| Net energy | MJ·ha$^{-1}$ | −10,172.7 | 46,693.1 | −57,374.6 | −6951.39 |
| Energy ratio | | 0.87 | 1.42 | 0.40 | 0.90 |
| Energy profitability | | −0.13 | 0.42 | −0.60 | −0.10 |
| Direct energy | MJ·ha$^{-1}$ | 25,984 | 24,581 | 30,203 | 26,923 |
| Indirect energy | kg·ha$^{-1}$ | 50,218.0 | 87,696.1 | 65,859.0 | 67,924.36 |
| Renewable energy | kg·MJ$^{-1}$ | 12,142.4 | 31,806.4 | 14,231.8 | 19,393.53 |
| Non-renewable energy | MJ·kg$^{-1}$ | 64,059.9 | 80,470.5 | 81,830.1 | 75,453.50 |

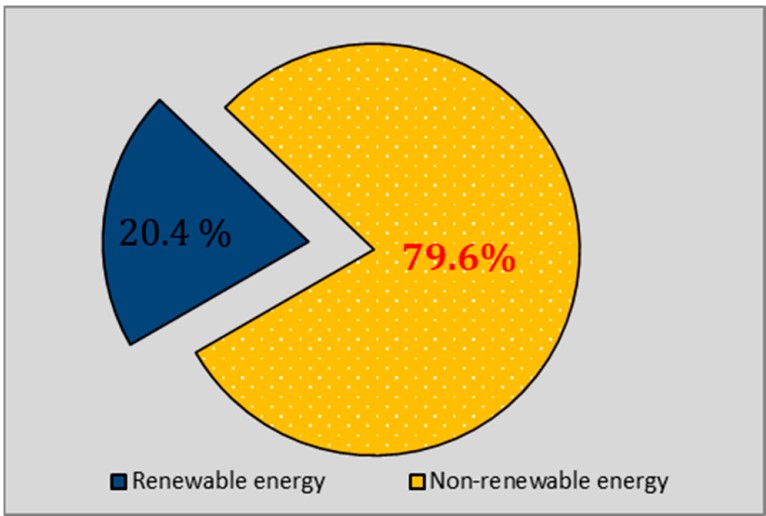

**Figure 4.** Comparison between the main contributions of renewable and non-renewable consumed energy in the production of vegetable crops.

### 3.3. Greenhouse Gas Emissions

Global warming is a significant struggle in the 21st century and represents one of the significant environmental challenges in the future. The continuing rise in the atmosphere and ocean temperatures was caused by increased concentrations of greenhouse gases ensuing from human activities such as deforestation and the burning of fossil fuels. Under the low GHG emission scenarios, the sea level will rise in the present century by 0.28–0.55 m [61]. Global warming exhibits the GHG contribution to climate change [62]. Agricultural GHGs ranged between 10–12% of total GHG emissions [63]. Total GHG emissions in vegetable crops in southern Egypt were determined using carbon emission

coefficients for each consumed indicator. The amount of each input is multiplied by their coefficient of GHG emissions to calculate the total GHG emissions.

GHG results of each crop are shown in Table 4. Onion produced the lowest rate of GHG emissions (3143 kg $CO_2$ eq.) per hectare, while tomato produced the highest amount of GHG emission (3742 kg $CO_2$ eq.) compared to both onion and potato. In onion fields, the highest rate of GHG emissions belonged to manure with a share of 29.7 of total emissions, followed by diesel (27%). Application of mineral nitrogen generates 70% of total mineral fertilization emissions in all cultivation processes. In potato farms, fertilizers (organic and inorganic) share more than 45% of total GHGs. As shown in Table 4, diesel and inorganic fertilizers produced 52% of the total GHG emissions of tomato fields. The main portion of GHGs was generated from fuel consumption [64,65]. GHG emissions of cultivation processes in southern Egypt varied from less than 200 kg $CO_2$ eq. $ha^{-1}$ in human labor, phosphorus and potassium and ranged between 200–700 kg $CO_2$ eq. $ha^{-1}$ in machinery, nitrogen and chemicals as a medium footprint, while diesel and manure produced more than 700 kg $CO_2$ eq. $ha^{-1}$.

**Table 4.** The amount of GHG (kg $CO_2$ eq. $ha^{-1}$) emissions concerning production inputs.

| Inputs | Onion | % | Potato | % | Tomato | % |
|---|---|---|---|---|---|---|
| Human labor | 137.9 | 4.4 | 134.7 | 4.0 | 162.2 | 4.3 |
| Machinery | 362.4 | 11.5 | 321.4 | 9.6 | 391.7 | 10.5 |
| Diesel | 1011.2 | 32.2 | 903.6 | 26.9 | 1165.8 | 31.2 |
| Nitrogen | 494.7 | 15.7 | 748.2 | 22.3 | 667.9 | 17.8 |
| Phosphorus | 45.1 | 1.4 | 62.3 | 1.9 | 56.8 | 1.5 |
| Potassium | 19.4 | 0.6 | 24.8 | 0.7 | 40.0 | 1.1 |
| Manure | 869.4 | 27.7 | 996.5 | 29.7 | 1009.1 | 27.0 |
| Chemicals | 202.9 | 6.5 | 168.8 | 5.0 | 248.3 | 6.6 |
| Total GHGs | 3142.9 | | 3360.3 | | 3741.9 | |
| GHGs ratio kg $CO_2$ eq. $ton^{-1}$ | 88.0 | | 76.0 | | 77.3 | |

Nitrogen fertilization produced the highest portion of GHG emissions compared to phosphorus and potassium [66]. GHG emission ratio measures the amount of $CO_2$ eq. generated by producing one ton of material. In our case, the production of one ton of potato, tomato and onion emitted 76, 88 and 77.3 kg $CO_2$ eq., respectively.

*3.4. Life Cycle Assessment (LCA)*

System boundary identity means setting criteria to define which unit processes are part of a product system. The selected system boundaries must be dependent on the aim of the life cycle study. The system boundaries are specified to include the activities contributing to the environmental consequences, regardless of whether they are inside or outside the cradle-to-grave system of the product [67]. The system boundary used in this study includes field preparation, crop management, crops product, pesticides and fertilizers.

The summarized results of the studied impact categories are presented in Table 5. LCA results provide comprehensive information on the environmental impacts of vegetable production on the ecosystem. GW impact generated from vegetable cultivation systems in southern Egypt was estimated to be between 238 kg $CO_2$ eq. in potato, as the lowest impact, to 283 kg $CO_2$ eq. $t^{-1}$ from onion farms, as the highest obtained GW value. For example, GW's impact on onion in Iran was 324 kg $CO_2$ eq. $t^{-1}$ [20]. Our research study indicated that agricultural machinery had the highest contribution to GW, with 45% of total emissions in onion and potato and 53% in tomato (Figure 5). Diesel contributes 17.5 % in potato and tomato and 23% in onion production. A similar study showed that agricultural machinery was the primary source of $CO_2$ emissions [68].

**Table 5.** Life cycle impact indicators per ton of crop product.

| Indicator | Unit | Onion | Potato | Tomato |
|---|---|---|---|---|
| Abiotic depletion (AD) | kg Sb eq. | 0.006 | 0.005 | 0.006 |
| Acidification (AC) | kg SO$_2$ eq. | 1.65 | 1.38 | 1.64 |
| Eutrophication (EU) | kg PO$_4$— eq. | 0.66 | 0.56 | 0.67 |
| Fresh water aquatic ecotoxicity (FAEF) | kg 1,4-DB eq. | 190.1 | 160.39 | 198.59 |
| Global warming (GW) | kg CO$_2$ eq. | 283.0 | 238.80 | 282.43 |
| Human toxicity (HT) | kg 1,4-DB eq. | 342.5 | 288.31 | 363.11 |
| Marine aquatic ecotoxicity (MAET) | kg 1,4-DB eq. | 433,226 | 365,636 | 454,463 |
| Ozone layer depletion (OLD) | kg CFC-11 eq. | 0.00006 | 0.00005 | 0.00006 |
| Photochemical oxidation (PO) | kg C$_2$H$_4$ eq. | 0.1 | 0.09 | 0.11 |
| Terrestrial ecotoxicity (TE) | kg 1,4-DB eq. | 1.35 | 1.18 | 1.36 |

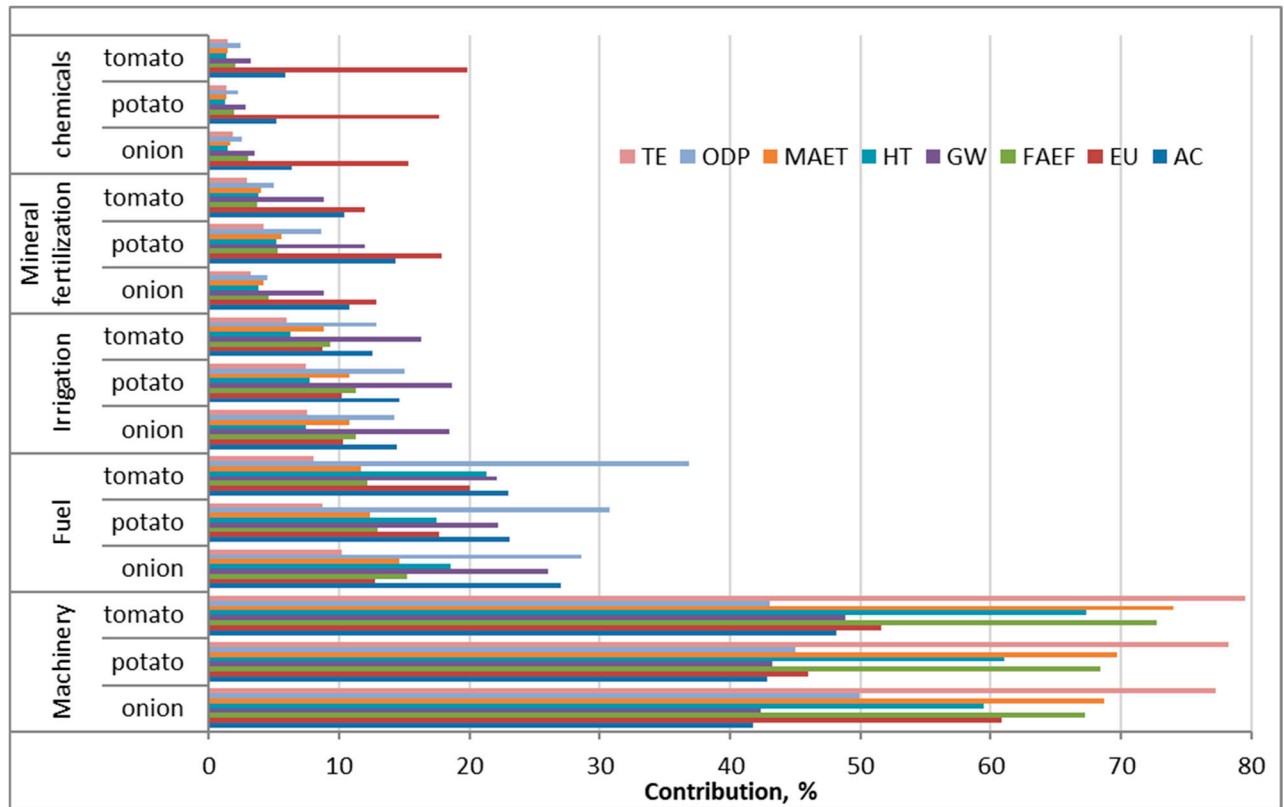

**Figure 5.** The contribution of individual inputs on the studied impact categories.

Brentrup et al. [62] indicated that HT, FAEF, TE and MAET are toxic categories that are harming the ecosystem. As set out, vegetable production emissions ranged from 288–363 kg 1,4-dichlorobenzene eq. per ton as the HT category. With more than 62%, agricultural machinery had the most significant impact. Previous studies refer to larger amounts observed in the values of HT equivalent to 1092 kg of 1,4-DCB per ton of wheat [69,70], and were 484 and 215 kg of 1,4-DCB per ton of olive [71,72].

Agricultural machinery, with a contribution of 64.8 and 78.6%, had the highest impact on FAEF and TE, followed by diesel fuel with 12 and 8.1%, respectively. FAEF and MAET of tomato production registered the highest values (189.6 and 433,487 kg 1,4-DB eq) per ton compared to onion and potato production. Onion production had the highest TE by 1.33 kg 1,4-DB eq per ton compared to 1.16 kg 1,4-DB eq per ton in potato fields. Comparing the results on a global scale, vegetable production in southern Egypt was more environmentally friendly in toxicity impact categories. In contrast, globally, as reported in the ecoinvent database [26], onion production resulted in 221, 571,000, 137 and 10.1 kg 1.4-DB in the impact categories of HT, MAET FAEF and TE, respectively [26,46].

Abiotic resources depleted 0.0055 kg Sb eq per ton of vegetables. Agricultural machinery, with 38% in onion and potato and 47% in tomato, had the most considerable impact in this category. AC, EU and PO of onion production generated the highest results compared to potato and tomato, producing 1.58 kg $SO_2$ eq, 0.56 kg $PO_4-q$ and 0.109 kg $C_2H_4$ eq, respectively. Acidification impact is mainly due to the release of $SO_2$, $NO_2$ and $NH_3$ into the air [62]. Khoshnevisan et al. 2014 [73] noticed that the production of one ton of cucumber and tomato cultivated in a greenhouse generates 0.63 and 0.37 kg $SO_2$ eq. EU potential for potato production was 1.0 kg $PO_4$ 3-eq in England [74]. Our results are still lower than the impact of onion production in the ecoinvent database, which registered 3.21 kg $PO_4$ [26], indicating that vegetable production in southern Egypt was environmentally friendly in this category. The obtained data for ozone layer depletion (OLD) showed an impact of 0.000055 kg CFC-11 eq per ton of vegetable product, which caused ozone depletion. Agricultural machinery, diesel and chemicals had the highest share of ozone-depleting pollutants. Production of one ton of tomato emitted $1.01 \times 10^{-6}$ kg CFC-11 eq to the ecosystem [46].

## 4. Conclusions

Onion, potato and tomato are considered essential cash crops in Egypt that generate high income, representing the main vegetable cultivated crops in the country.

The study aimed to evaluate the energy balance and potential environmental impact of the main vegetable crop production processes in southern Egypt. The results showed that the generated energy is more than the consumed energy in potato fields (140%) and vice versa in onion and tomato farms (87 and 40%, respectively). The highest percentages of energy input came from inorganic fertilizers, followed by diesel, water, chemicals and machinery. Onion production emitted 3143 kg $CO_2$ eq. $ha^{-1}$ while tomato produced the highest amount of GHGs, recording 3742 kg $CO_2$ eq. $ha^{-1}$. Reducing diesel fuel, chemical fertilizer and irrigation water (and seeds for potatoes) are the most crucial methods for improving the energy management of agricultural systems in the study region. This goal recommends determining soil fertility to optimize chemical fertilizer application, reducing GHG emissions and potential environmental impacts. It is also vital to integrate pest management to reduce chemical usage and reduce abiotic resource depletion. The transformation to modern irrigation methods instead of traditional methods, such as furrows and floods, will reduce water loss, increase water use efficiency and decrease FAEF and marine MAET in arid zones.

**Author Contributions:** Manuscript conception, methodology, data analysis, validation, investigation, writing—original draft preparation, M.A. and M.Z.; writing—review and editing, K.M.-S.M.; project administration, M.A. and M.R.N. All authors have read and agreed to the published version of the manuscript.

**Funding:** This paper was supported by the Kadyrov Chechen State University's development program 2021–2030.

**Institutional Review Board Statement:** Not applicable.

**Informed Consent Statement:** Not applicable.

**Data Availability Statement:** The data presented in this study are available in the article.

**Acknowledgments:** This paper was supported by the Kadyrov Chechen State University development program 2021–2030.

**Conflicts of Interest:** The authors declare no conflict of interest.

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
