# Peer review of "Life Cycle Assessment of the Cultivation Processes for the Main Vegetable Crops in Southern Egypt"

_agronomy, doi:10.3390/agronomy12071527_

Round 1

Reviewer 1 Report

The manuscript is well written. There are some minor revisions before the acceptance. 

1. Line 288, why seeds energy ranked secondly in potato cropping system compare to other crops such as tomato and onion? Please give more details. 

2. Line 283, potaton should be onion.

3. Line 287, 404, Onion and Production should be onion and production. 

4. Line 417 OLD (Ozone layer depletion) can be expressed as  "Ozone layer depletion (OLD)".

Author Response

Dear professor

We want to thank you for your thoughtful comments and efforts toward improving our manuscript. In the following text, we listed your comments (Bold) and our answers

  • Line 288, why seeds energy ranked secondly in potato cropping system compare to other crops such as tomato and onion? Please give more details. 

Potato seeds (called True potato seeds) refer to potato tuber used for cultivation processes (vegetative reproduction). Potato cultivation needs a massive amount of potato tubers per hectare. In addition, one kg of potato seeds (tuber) produces 3.6 MJ as the energy equivalent. By multiplying these numbers, a great amount of energy will be produced (representing 17% of the potato cultivation process). Vice versa in tomato and Onion cultivation, whereas sexual seeds are used for cultivation. One kg of these seeds (onion and tomato) produces just one MJ as energy equivalent.

Line 283, potato should be onion. Have been corrected

Line 287, 404, Onion and Production should be onion and production.

Have been corrected

Line 417 OLD (Ozone layer depletion) can be expressed as  "Ozone layer depletion (OLD)".

Have been corrected

Sincerely,

Dr. Mostafa

Reviewer 2 Report

As I stated in the previous round of revision (although only directly to the Editors, and not through the online form), I have some concerns about the GHG emission coefficients used. For example, the coefficient for N fertiliser is listed as 1.3 kg CO2 eq. kg−1. I would expect this to be closer to 6 kg CO2 eq. kg−1, if the figure corresponds to emissions per kg N and not per kg of fertiliser (which contains more than just N). Please see for example the Standard Values used in the Biograce tool available at BIOGRACE. It may be that in fact total chemical weight was used, which would also explain the very high N application rates listed for the crops (up to 576 kg per ha for potato - which is more than double what I would have expected). However, this is not clear.

May I please request that the authors look over their GHG emission coefficients once more, confirm that they are indeed correct for the inputs applied, and clarify exactly what the unit of each input means (e.g. active ingredient or total weight).

Author Response

Dear professor

We want to thank you for your thoughtful comments and efforts toward improving our manuscript.  We tried to answer your comment in the following text (your comment in the BOLD line)

  • As I stated in the previous round of revision (although only directly to the Editors, and not through the online form), I have some concerns about the GHG emission coefficients used. For example, the coefficient for N fertiliser is listed as 1.3 kg CO2 eq. kg−1. I would expect this to be closer to 6 kg CO2 eq. kg−1, if the figure corresponds to emissions per kg N and not per kg of fertiliser (which contains more than just N). Please see for example the Standard Values used in the Biograce tool available at BIOGRACE. It may be that in fact total chemical weight was used, which would also explain the very high N application rates listed for the crops (up to 576 kg per ha for potato - which is more than double what I would have expected). However, this is not clear. May I please request that the authors look over their GHG emission coefficients once more, confirm that they are indeed correct for the inputs applied, and clarify exactly what the unit of each input means (e.g. active ingredient or total weight).

The applied coefficient for the total weight of fertilizers in our work is according to (Ilahi et al., 2019) who applied 1.3 CO2 eq. kg−1 coefficients for nitrogen fertilizers. Also, our work is in harmony with Lal, 2004 (Carbon emission from farm operations, 2008) which refers to the acceptable coefficient for nitrogen fertilizer between 0.9 – 1.8 CO2 eq. kg−1. The report of ‘BioGrace-I calculation rules Version 4d (page 7-Table 1) refers to The highest value for Nitrogen fertilisers emissions is 5.88 CO2 eq per kg of nitrogen as an active component not as nitrogen fertilisers (It will be about 1.9 CO2 eq per kg of nitrogen if ammonium nitrate used - 33.5% N). Also, the report refers to the highest standard coefficients applied in specific cases mentioned in the report.
•    The point of the total weight of applied fertilizer was cleared in the manuscript (line- 258, directly after figure 2 in the text).

Sincerely,

Dr. Mostafa